# CCC: Prompt Evolution for Video Generation via Structured MLLM Feedback

## Abstract

Video generation from natural-language prompts has made impressive strides, but current systems frequently misalign outputs with their input descriptions—dropping critical details, hallucinating unintended content, and often violating basic physical plausibility. Existing approaches to improving video quality typically rely on heavyweight post-editing models, which may introduce new artifacts, or costly fine-tuning of the generator backbone, limiting scalability and accessibility. We introduce `Critique Coach Calibration` (CCC), a training-free, model-agnostic framework for iterative prompt evolution that closes the gap between direct generation and reality. In each cycle, an off-the-shelf multimodal large language model (MLLM) produces a structured critique of a generated video—identifying semantic misalignments, subject drift, and missing objects—and then reformulates the input prompt based on this feedback. By repeatedly evolving prompts through this critique–coach loop, CCC steadily improves both video quality and adherence to everyday physics, all without modifying the generator or relying on external editing modules. Empirical results on different video models show that CCC consistently enhances semantic alignment and visual quality through the power of adaptive prompt evolution.

## 1 Introduction

**Prompt:** The brightly painted skateboard, adorned with bold graffiti designs, glides along the sunlit sidewalk; nearby, kids laugh and chase after it, their shadows stretching long in the golden afternoon light.

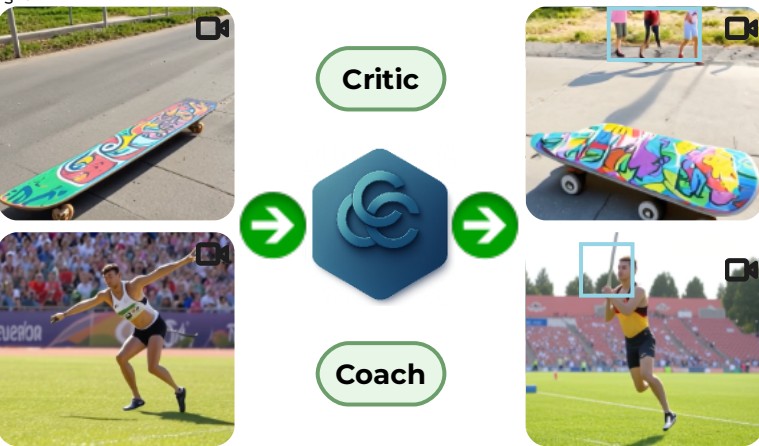

**Prompt:** Under the intense mid-afternoon sun, an elite Olympic athlete powerfully hurls a javelin across the vibrant green field, the projectile slicing through the air with mesmerizing grace.

Figure 1: **Two examples of `Critique Coach Calibration` (CCC)'s impact:** *Top:* The first generated video does not include the children, which is a key detail in the prompt. After CCC refines the prompt, the video on the right includes the children in the back. *Bottom:* In the first generated video, the athlete is not holding a javelin, which is a crucial detail of the prompt. After CCC refines the prompt, the video on the right includes the javelin.

Video generation from natural language has rapidly advanced, enabling the creation of rich, dynamic scenes from simple textual descriptions (Ho et al., 2022; Singer et al., 2023; Blattmann et al., 2023; Chen et al., 2024). Despite this progress, generated videos often suffer from a range of issues—including poor semantic alignment with the input prompt and subpar visual quality. Models may omit key details, hallucinate unintended content, or fail to maintain consistency across frames, reflecting a disconnect between the intended meaning and the visual output. These limitations reflect a broader misalignment between the intended meaning of the input prompt and the overall fidelity of the visual output—posing fundamental challenges for scalable and trustworthy video generation.

Prior approaches to video improvement generally fall into two categories: (1) post-hoc editing using auxiliary models (Lee et al., 2025; Huang et al., 2024), and (2) architectural modifications to the generative model itself (Ahn et al., 2024). However, both strategies are resource-intensive, as they often require substantial design effort and specific training samples, while video editing-based methods can introduce unwanted artifacts. Meanwhile, multimodal large language models (MLLMs) have demonstrated strong capabilities in textual reasoning and, when paired with visual inputs, can identify inconsistencies or mismatches in generated content (Liu et al., 2025; Xiong et al., 2024). This evaluative strength positions MLLMs as promising tools for diagnosing failures in visual content. Existing efforts to enhance visual quality have primarily focused on still images (Yang et al., 2024; Mañas et al., 2025; Singh & Zheng, 2023), whereas improving videos remains significantly more challenging due to their temporal dynamics and consistency requirements.

To address these challenges, we introduce `Critique Coach Calibration` (CCC), a novel framework designed to unlock the full potential of existing text-to-video models without altering their architecture or directly editing their outputs. Figure 1 illustrates two concrete examples of how CCC identifies missing details (e.g., children in a stadium scene, an absent javelin in an athletics clip) and then refines the prompt to correct those omissions. The framework operates in two stages: a critic stage, where an off-the-shelf multimodal LLM (MLLM) is calibrated to produce structured diagnostic interpretable reports on issues such as semantic misalignment, subject drift, and object absence; and a coach stage, where these critiques are fused with the original prompt, rewritten by the MLLM, and used to generate new videos. By iteratively alternating between these stages, CCC enables a self-improving generation loop that enhances both text-video alignment and overall visual quality without manual intervention or model-specific finetuning.

Our experiments on diverse video scenarios show that `Critique Coach Calibration` yields substantial gains in alignment fidelity and narrative coherence with human evaluation. Qualitative results further illustrate the superiority of our pipeline compared with baselines. Additionally, we conduct extensive ablation studies on the critic reliability, prompt refinement, and calibration strategies, which show the effectiveness of our designs.

Our main contributions are summarized as follows:

- We introduce `Critique Coach Calibration` (CCC), a training-free model-agnostic framework that can improve the general text-to-video generation model performance with fully interpretable intermediate steps.
- We design a video critic pipeline by leveraging pretrained MLLM for identifying multimodal misalignment and visual distortion during video generation.
- We conduct comprehensive evaluations on diverse video samples and show that CCC can provide higher quality videos than baselines. Further ablations demonstrate the effectiveness of our module design.

## 2 RELATED WORK

While a growing body of work has sought to improve text-to-video generation, most prior methods target a single failure mode whereas our framework addresses all of these jointly through a unified, model-agnostic, training-free refinement loop. For example, VideoRepair focuses exclusively on detecting and re-diffusing mis-aligned regions to correct object-level semantic errors (Lee et al., 2025), but it cannot adjust global prompt phrasing or handle temporal drift; in contrast, our Critique Coach Calibration (CCC) uses structured MLLM feedback to rewrite the entire prompt and fully re-generate the clip, capturing both object- and sequence-level issues.

## 2.1 MLLMs as Video Judges

Recent work repurposes multimodal large language models (MLLMs) as off-the-shelf evaluators for generated video. AIGV-Assessor (Wang et al., 2024) and LMM-VQA (Ge et al., 2024) reformulate video quality assessment as a VQA task; UVE (Liu et al., 2025), Evaluation Agent (Zhang et al., 2024), VideoAutoArena (Luo et al., 2025), and LLaVA-Critic (Xiong et al., 2024) demonstrate that prompted MLLMs can approximate human judgements across low- and high-level dimensions. However, these systems (i) use the MLLM solely for scoring or ranking, (ii) output a single scalar or ranking without actionable guidance, and (iii) do not leverage the same MLLM to directly reformulate prompts. By contrast, CCC "reuses" the very same MLLM as both *critic* and *coach*: it generates a structured, multi-dimensional critique of the prompt–video pair, then feeds that critique back into a second pass of the *same* MLLM to rewrite the prompt, enabling an evaluation-to-refinement loop without external modules or multiple models.

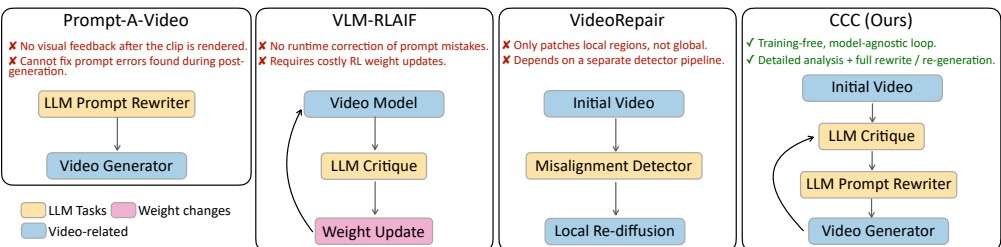

Figure 2: **Training-free, model-agnostic refinement loop.** Prompt-a-Video, VLM-RLAIF, and VideoRepair either lack visual feedback, require RL weight updates, or patch only local regions. Our loop uses an LLM critique to rewrite the *entire* prompt, then fully re-generates the clip with no parameter tuning.

## 2.2 Prompt Optimization and Iterative Repair

Existing methods inject feedback at different stages of the generation pipeline (see Figure 2): pre-generation prompt rewriting (Prompt-A-Video (Ji et al., 2024), VPO (Cheng et al., 2025)), training-time alignment (VLM-RLAIF (Ahn et al., 2024), DPO-Video (Zhang et al., 2025)), and post-generation local repair (VideoRepair (Lee et al., 2025), CSR (Zhou et al., 2024)). Each single-stage approach corrects only one aspect, either before rendering, during model training, or via local patching, but cannot address emergent, global failures across multiple dimensions. In contrast, CCC combines (i) post-generation MLLM critique, (ii) global prompt-level rewrite, and (iii) full re-generation in a multi-round loop. By accepting rewrites only when there is self-consistency (i.e., the Content Agreement Score is sufficient), CCC achieves reliable, global corrections without any weight updates, auxiliary detectors, or model-specific tricks, making it readily applicable to any text-to-video backbone.

## 2.3 Self-Consistency and Reliable Evaluation

Self-consistency decoding (Wang et al., 2023) improves LLM reasoning by majority voting over multiple sampled chains of thought, and recent work studies uncertainty in LLM judgements (Xie et al., 2025) to gauge evaluator reliability. These methods demonstrate that (i) outputs can vary significantly across runs and (ii) aggregating multiple runs yields more stable verdicts. In the visual domain, some systems use repeated prompting to reduce variability, but none integrate the consistency signal into a closed-loop refinement. CCC extends these ideas by (a) querying the MLLM $K$ times per prompt–video pair, (b) computing both a self-consistency score (majority agreement on overall rating) and a Content Agreement Score (fine-grained consensus on detected issues), and (c) conditioning prompt refinements on these reliability measures and doing so uniformly for both images and videos.

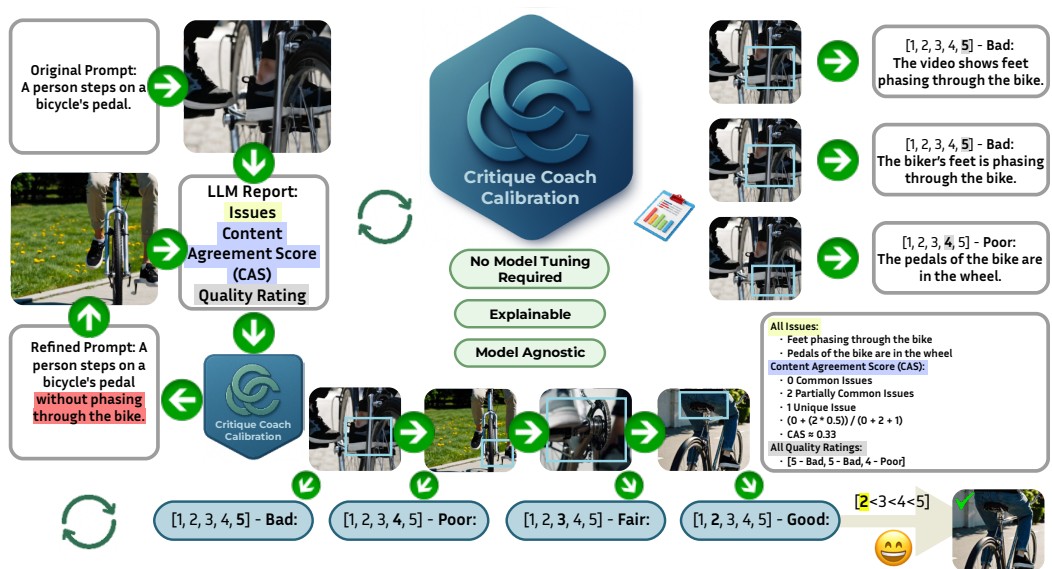

Figure 3: **Overview of `Critique Coach Calibration` (CCC):**. MLLM first issues K self-consistent critiques of a generated clip, aggregates recurring failure modes into severity tiers (e.g., *Major*, *Minor*), and then distills the dominant findings into calibrated guidance that steers the next generation cycle. The bounding boxes are for visualization only, not generated in our pipeline.

## 3 Critique Coach Calibration

We now describe our `Critique Coach Calibration` (CCC) pipeline, which wraps any frozen text-to-video generator into a self-diagnosing, self-correcting loop (Figure 3). At each iteration $t$, the input prompt $\mathcal{P}^{(t)}$ is sent to the generator $G_\theta$ to produce a candidate video clip $\mathcal{V}^{(t)} = G_\theta\big(\mathcal{P}^{(t)}\big)$. An off-the-shelf multimodal LLM then serves two roles:

- **Critic:** Given the pair $(\mathcal{P}^{(t)}, \mathcal{V}^{(t)})$, it issues a coarse quality rating $y^{(t)} \in \{1, \ldots, 5\}$ (1=Excellent...5=Bad) plus a set of textual findings $\mathcal{C}^{(t)} = \{\delta_1, \ldots, \delta_m\}$, where each $\delta_i$ names a single failure mode (e.g. "missing children in frame").

- **Coach:** It consumes $(\mathcal{P}^{(t)}, \mathcal{C}^{(t)})$ and outputs a corrective rewrite $\mathcal{R}^{(t)}$ that explicitly targets the identified flaws.

We then update the prompt via $\mathcal{P}^{(t+1)} = \mathcal{P}^{(t)} \oplus \mathcal{R}^{(t)}$ and repeat for $T$ rounds, progressively improving text-to-video alignment and visual fidelity without any model-specific tuning (Section 3.3).

### 3.1 RELIABILITY OF THE CRITIC

**Setup.** We extract $F$ evenly-spaced frames per video so as to sample both beginning and end positions. We generate $M$ prompt variants to cover diverse negative-prompting and phrasing strategies. Finally, we query an MLLM to capture model-agnostic feedback. For stability analysis, we further sample the same LLM $K$ times with identical inputs.

**Quality-rating consistency.** Across the $K$ independent runs we obtain a set of quality ratings $\{y_k\}_{k=1}^K$, where each $y_k \in \{1, \ldots, 5\}$ denotes the critic's coarse perceptual score (1=Excellent, ..., 5=Bad). We define $\text{mode}(y)$ to be the most frequently occurring rating among the $y_k$. The consistency score $\gamma \in [0, 1]$ then measures the fraction of runs that agree with this majority rating:

$$\gamma = \frac{\big|\{\, k : y_k = \text{mode}(y)\}\big|}{K} \tag{1}$$

A high $\gamma$ indicates that the critic is reliably assigning the same overall quality judgment.

---

**Algorithm 1** CCC Loop (cf. Figure 3)

---

1: **Input:** Initial prompt $\mathcal{P}^{(0)}$, generator $G_\theta$, critic/coach LLM $\phi$, iterations $T$, repetitions $K$, samples per round $M$
2: **for** $t = 0$ to $T - 1$ **do**
3:     Sample candidate videos $\{V_i^{(t)}\}_{i=1}^M \leftarrow G_\theta(\mathcal{P}^{(t)})$ ▷ Generate initial candidate videos
4:     **for** $i = 1$ to $M$ **do**
5:         Query $\{(y_{i,k}, \mathcal{C}_{i,k})\}_{k=1}^K$ from $\phi$ ▷ Obtain $K$ critic /score pairs
6:         Compute $\gamma_i$ via Eq. equation 1 and $\Psi_{\text{CAS},i}$ via Eq. equation 3 ▷ Compute confidence and CAS metrics

7:         Select canonical critique $\mathcal{C}_i^{(t)}$ with highest $\Psi_{\text{CAS},i}$ ▷ Keep critique with highest CAS
8:     **end for**
9:     $i^* \leftarrow \arg\max_i \Psi_{\text{CAS},i}$ ▷ Choose clip with best CAS
10:     $\mathcal{C}^{(t)} \leftarrow \mathcal{C}_{i^*}^{(t)}$ ▷ Use its critique as canonical
11:     $\mathcal{R}^{(t)} \leftarrow f_\phi(\mathcal{P}^{(t)}, \mathcal{C}^{(t)})$ ▷ Produce rewrite via critic
12:     $\mathcal{P}' \leftarrow \mathcal{P}^{(t)} \oplus \mathcal{R}^{(t)}$ ▷ Augment prompt with rewrite
13:     Sample refined videos $\{V_j'^{(t)}\}_{j=1}^M \leftarrow G_\theta(\mathcal{P}')$ ▷ Generate refined videos
14:     **for** $j = 1$ to $M$ **do**
15:         Rate each refined clip yielding $y_j'$ ▷ Critic rates each refined clip
16:     **end for**
17:     $j^* \leftarrow \arg\min_j y_j'$ ▷ Select best refined clip
18:     $V^{(t)} \leftarrow V_{j^*}'^{(t)}$ ▷ Save chosen clip
19:     $\mathcal{P}^{(t+1)} \leftarrow \mathcal{P}'$ ▷ Set prompt for next iteration
20: **end for**
21: **Output:** Final prompt $\mathcal{P}^{(T)}$ and chosen videos $\{V^{(t)}\}_{t=0}^{T-1}$

---

**Content Agreement Score.** Within those $K$ runs, each produces a set of textual issues $\mathcal{I}_k$ (e.g. "missing object X," "subject drift"). To quantify how consistently the critic identifies the same low-level flaws, we compute three counts:

$$A = \Big|\bigcap_{k=1}^K \mathcal{I}_k\Big|, \qquad B = \Big|\bigcup_{k=1}^K \mathcal{I}_k\Big| - \Big|\bigcap_{k=1}^K \mathcal{I}_k\Big|, \qquad C = \sum_{k=1}^K |\mathcal{I}_k| - A - B. \qquad (2)$$

Here: - $A$ is the number of issues cited by all $K$ runs, - $B$ is the number cited by some but not all runs, - $C$ is the total count of issues that appeared in exactly one run (i.e. unique issues).

We then define the Content Agreement Score $\Psi_{\text{CAS}} \in [0, 1]$ as

$$\Psi_{\text{CAS}} = \frac{A + 0.5\,B}{A + B + C} \qquad (3)$$

A value of $\Psi_{\text{CAS}} = 1$ means every run extracted exactly the same set of issues, indicating maximal diagnostic consistency.

## 3.2 TURNING FEEDBACK INTO A BETTER PROMPT

We cast the coaching step as a two-stage *chain-of-thought* dialogue with the MLLM. **(1) Issue discovery.** Given the generated video and its current prompt, the MLLM first lists concrete problems it detects (e.g., missing objects, semantic drift, or temporal artifacts). **(2) Prompt refinement.** We then supply both the *previous* prompt and this issue list to the same MLLM, instructing it to produce an improved prompt that resolves the highlighted flaws *without omitting any correct details* from the original. By iteratively applying this conservative refinement, the system corrects errors while avoiding fresh hallucinations, yielding steadily better generations across rounds.

## 3.3 FULL CCC LOOP

Figure 3 and Algorithm 1 together depict the end-to-end CCC pipeline. At iteration $t$, we first sample $M$ candidate videos $\{V_i^{(t)}\}_{i=1}^M$ from the frozen generator $G_\theta$ using the current prompt $\mathcal{P}^{(t)}$. For each candidate $V_i^{(t)}$, we invoke the Critic–Coach LLM $\phi$ $K$ times to obtain critiques $\{(y_{i,k}, \mathcal{C}_{i,k})\}_{k=1}^K$.

We then compute the consistency score $\gamma_i$ via Eq. 1 and the content agreement score $\Psi_{\text{CAS},i}$ via Eq. 3, and select the canonical critique $\mathcal{C}_i^{(t)}$ corresponding to the run that maximizes $\Psi_{\text{CAS},i}$.

$$i^* = \arg\max_i \Psi_{\text{CAS},i} \tag{4}$$

We refine the current prompt by appending the corrective rewrite suggested by the coach, then select the highest-quality video among the new candidates. Formally:

$$\mathcal{P}' = \mathcal{P}^{(t)} \oplus \mathcal{R}^{(t)} \tag{5}$$

$$j^* = \arg\min_j y_j' \text{ (best quality clip)} \tag{6}$$

Here, $\mathcal{P}^{(t)}$ is the prompt at iteration $t$, $\mathcal{R}^{(t)}$ is the refinement produced from the canonical critique, and $\oplus$ denotes concatenation of the original and rewritten text. We sample $M$ candidate videos from the generator using the updated prompt $\mathcal{P}'$, score each clip with the critic to obtain $y_j'$, and select the best-rated result $V^{(t)} = V_{j^*}'^{(t)}$. The chosen video and updated prompt then initialize the next iteration, continuing the refinement loop.

# 4 EXPERIMENTAL RESULTS

## 4.1 SETTING

**Implementation details.** We use **o1 as the main feedback MLLM**. The MLLM generates textual responses for every prompt–video pair, forming a comprehensive output matrix. To measure prompt effectiveness, we combine Amazon Turk and manual evaluation, revealing how wording and context influence MLLM video analysis and informing automated prompt refinement. As for the **testing video models**, we used LTX-Video (HaCohen et al., 2024) and WanX-2.1 (Wan et al., 2025), two widely used open-source SOTA models. We excluded proprietary models since their APIs often apply hidden prompt rewriting, which could alter our refined prompts.

For **baselines**, *Origin* forwards the raw prompt $P^{(0)}$ to $G_\theta$ and selects the generated clip, providing a no-feedback reference. *LLM-P* (LLM Paraphrasing) performs a single text-only paraphrase: GPT-4o rewrites $P^{(0)}$ *without* seeing any video, after which $M$ new clips are generated and scored. *OPT2I* (Mañas et al., 2025) executes one round of visual feedback: GPT-4o first paraphrases $P^{(0)}$, inspects a draft clip, issues a second rewrite $\hat{P}$, and we regenerate $M$ clips from $G_\theta(\hat{P})$.

Table 1: **Human evaluation results on both semantic alignment and visual quality.** Percent of raters preferring our model over baselines. **SA** = semantic alignment, **VQ** = video quality, **-N** = without negative strategy. Numbers in parentheses denote change ($\Delta$) relative to the "-N" baseline.

| | LTX-Video | | | Wanx-2.1 | | |
|---|---|---|---|---|---|---|
| | Origin | OPT2I | LLM-P | Origin | OPT2I | LLM-P |
| **SA-N** | 61.4% | 55.7% | 56.6% | 59.7% | 62.9% | 63.4% |
| **VQ-N** | 51.7% | 65.7% | 55.4% | 57.1% | 58.0% | 55.4% |
| **SA** | 62.8% ($\Delta$+1.4%) | 57.1% ($\Delta$+1.4%) | 62.9% ($\Delta$+6.3%) | 62.3% ($\Delta$+2.6%) | 60.4% ($\Delta$–2.5%) | 65.6% ($\Delta$+2.2%) |
| **VQ** | 54.5% ($\Delta$+2.8%) | 64.0% ($\Delta$–1.7%) | 57.7% ($\Delta$+2.3%) | 61.8% ($\Delta$+5.7%) | 59.6% ($\Delta$+1.6%) | 58.5% ($\Delta$+3.1%) |

As for the **metrics** to evaluate the quality of generated videos, we combine both **automatic semantic alignment** metrics and **human visual preference** judgments. For semantic alignment, we follow the evaluation protocol of EvalCrafter (Liu et al., 2024). Specifically, we use three complementary signals: (1) Text–Video Consistency (CLIP-Score), measuring cosine similarity between prompt text and video frames using CLIP ViT-B/32; (2) Image–Video Consistency (SD-Score), comparing video frames with SDXL-generated images to reduce concept forgetting; and (3) Text–Text Consistency (BLIP-BLEU), comparing BLIP2-generated captions with the original prompt via BLEU. These automatic metrics track semantic fidelity but cannot fully capture perceptual realism and aesthetic quality. Therefore, we also conduct a **human evaluation** to assess both **visual quality** and **semantic alignment**: annotators are shown side-by-side video pairs and asked to vote for the one with corresponding metrics.

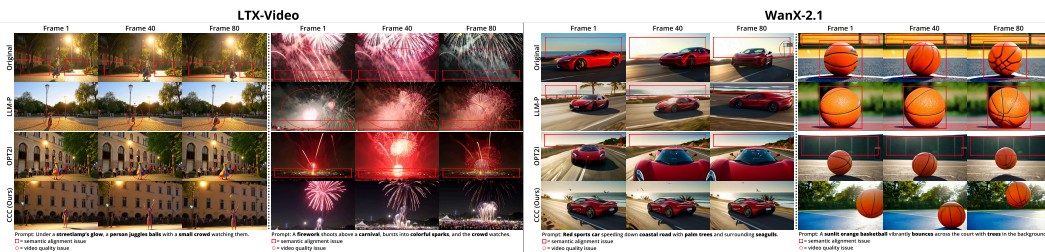

Figure 4: **Qualitative Comparison with Baselines.** We showcase four illustrative cases—two produced with LTX-Video and two with Wanx-2.1. While the baselines strive to enhance video quality, CCC yields markedly superior generations.

In our main experiments we employ $T = 3$ calibration rounds. Video generation requires on average 15 seconds per video when using the LTX model and 120 seconds per video when using the WANX (Wan et al., 2025) model. As each round includes one generation step, the total wall-clock time per video–prompt pair is therefore approximately 45 s under LTX and 360 s under WANX.

## 4.2 MAIN RESULTS

Table 2: **Automatic evaluation results.** Critique Coach Calibration boosts the performance of LTX-Video and Wanx-2.1 on semantic alignment.

| Method | Origin | LLM-P | OPT2I | CCC (ours) |
|---|---|---|---|---|
| LTX-Video (HaCohen et al., 2024) | 63.1 | 63.6 | 64.5 | **68.9** |
| Wanx-2.1 (Wan et al., 2025) | 68.1 | 68.5 | 67.7 | **70.3** |

Text-to-video evaluation spans two key dimensions: **(i) Semantic alignment** — how faithfully the generated clip follows its textual prompt — and **(ii) Perceptual quality** — the visual realism and overall watchability of the video.

Because perceptual quality is difficult to measure automatically, we rely on human preference studies summarized in Table 1. Annotators compared our CCC outputs against three training-free baselines (Origin, OPT2I, and LLM-P) across two widely used backbones, LTX-Video and Wanx-2.1. For **LTX-Video**, CCC improves over the no-negative baseline: semantic alignment rises from 61.4% to 62.8% and perceptual quality from 51.7% to 54.5%. CCC is competitive or better — e.g., SA 62.8% vs. 57.1% (OPT2I) and 62.9% (LLM-P); VQ 54.5% vs. 64.0% (OPT2I) and 57.7% (LLM-P). For **Wanx-2.1**, we also observe clear improvements: SA increases from 59.7% to 62.3% and VQ from 57.1% to 61.8% relative to the no-negative baseline, while maintaining competitive performance against OPT2I (60.4% SA, 59.6% VQ) and LLM-P (65.6% SA, 58.5% VQ).

Complementing human evaluation, Table 2 shows that CCC also yields higher automatic semantic alignment scores, reaching 68.9 on LTX-Video and 70.3 on Wanx-2.1, outperforming all baselines.These results confirm that our iterative critique–refine loop improves both semantic fidelity and perceived visual quality across multiple backbones. The qualitative result in Figure 4 echoes these findings. Across four representative prompts, Critique Coach Calibration removes the red-boxed semantic mismatches and red-circled visual artifacts that linger in competing methods, underscoring its strength in prompt-level refinement and overall video quality.

## 4.3 EFFECTIVENESS OF NEGATIVE STRATEGY

In our experiments, to better finding the issues in the video, to utilize the the "negative strategy" to assess the impact of explicitly warning the MLLM about potential errors versus a neutral direct prompt. We first conduct a controlled evaluation against human judgments. For each video, we compare two prompting conditions as in Figure 5. As illustrated in the two examples, when adding a fault- seeking clause ("There could be issues. . . ") draws richer and more accurate critiques from the MLLM. Across two generated videos, the negative- prompt strategy surfaces missing details that

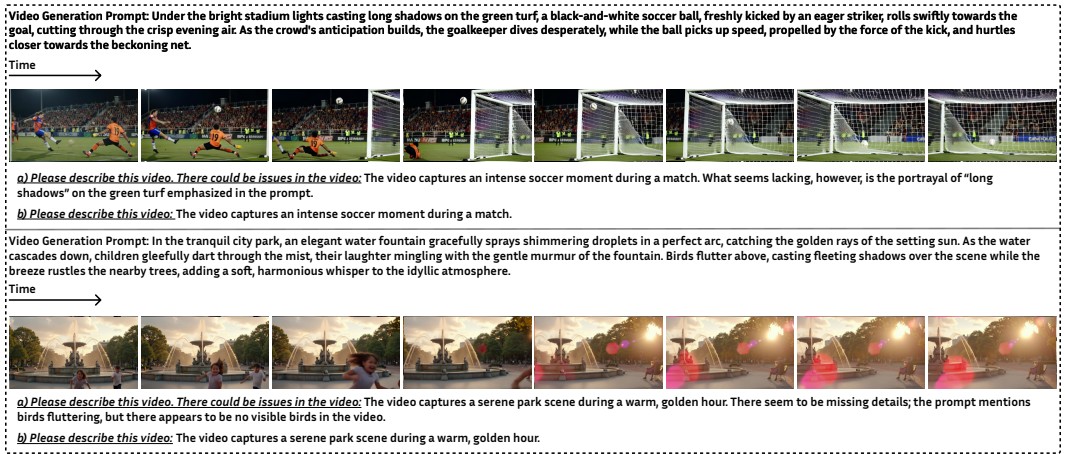

Figure 5: **Adding a fault-seeking clause ("There could be issues...") draws richer and more accurate critiques from the MLLM.** Across two generated videos, the negative-prompt strategy surfaces missing details that the neutral direct prompt overlooks, shifting responses closer to human ground truth.

the neutral direct prompt overlooks, shifting responses closer to human ground truth when asking about what are the issues in the generated videos as the MLLM prompts in Appx. B.

Table 3: Ablation results: effect of removing key components in the prompt optimization process on **visual quality**. It shows win rate of CCC over each ablation.

| Ablation Variant | Alignment with Human |
|---|---|
| W/ Both | 50.0% |
| No Self-Consistency | 45.0% |
| No Iteration Loop | 38.5% |
| No Both | 35.5% |

Table 4: Ablation results: effect of removing key components on **semantic alignment** using the automatic metric. Higher is better.

| Ablation Variant | Semantic Alignment |
|---|---|
| W/ Both | 70.3 |
| No Self-Consistency | 68.5 |
| No Iteration Loop | 69.1 |
| No Both | 68.1 |

## 4.4 ABLATION STUDY

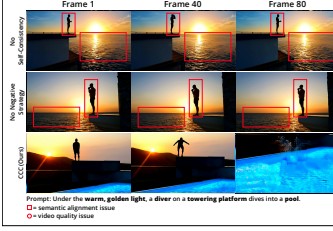

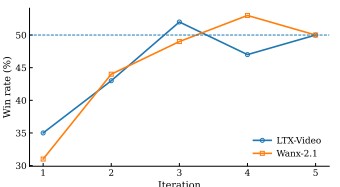

Figure 6: **Comparison with Ablations.** Critique Coach Calibration keeps the subject, motion, and scene intact, whereas ablations miss actions or key objects.

Figure 7: **Iteration-wise scores on semantic alignment.** Performance across iterations for LTX-Video and Wanx-2.1.

Figure 8: **Human Preference vs Iter-5. on visual quality.** Win rate when each iteration is compared to iteration 5 (50% = tie).

We conduct three ablations to isolate the impact of key components in our pipeline. In the *No Self-Consistency* variant, we skip the repeated sampling and content agreement analysis, using only the first MLLM response for prompt refinement. In the *No Negative Strategy* variant, we omit the initial error-warning phrasing and rely on a neutral direct prompt for evaluation. In the *No Iterative*

*Loop* variant, we perform only one round of evaluation and refinement rather than iterating the feedback cycle. This comparison reveals how much each component—self-consistency, negative strategy, and iterative looping—contributes to the final video quality.

We conduct a more detailed investigation into the influence of different components within our prompt refinement framework. Specifically, we analyze the effects of removing key modules through an ablation study, as summarized in Tables 3 and 4. We consider two ablated variants: (1) *No Self-Consistency*, where the model receives only a single critique per refinement cycle instead of aggregating multiple feedbacks, and (2) *No Iteration Loop*, where the prompt is refined only once without iterative refinement. Our results indicate that both components are essential for achieving high-quality video generation. However, the absence of iterative refinement (*No Iteration Loop*) results in a more pronounced drop in performance, highlighting its critical role in progressively guiding the generation model toward improved alignment and fidelity.

To illustrate these differences, Figure 6 shows qualitative results for a sample prompt: "Under the warm, golden light, a diver on a towering platform dives into a pool." We visualize four key frames from each variant to compare temporal consistency and semantic accuracy. Baselines such as No Negative Strategy and No Self-Consistency often show subject drift, disappearing elements, and visual artifacts. In contrast, our full CCC preserves subject integrity, coherent motion, and scene structure, highlighting the value of iterative feedback and consistency mechanisms for faithful, high-quality video generation.

We further conducted experiments to evaluate how iterative refinement affects both **semantic alignment** and **visual quality**. As shown in Figure 7, the middle plot reports semantic alignment using an automatic metric, while Figure 8 on the right measures visual quality through pairwise human voting against iteration 5 (with 50% indicating no preference). Because perceptual realism is challenging to capture with a single automatic model, we rely on human judgments for the right-hand analysis. Across both metrics, performance generally peaks around iteration 3; additional refinement does not consistently improve results and can even degrade them. We attribute this to the feedback model drifting away from the original task goal after multiple rounds, gradually losing essential prompt factors and introducing unintended changes.

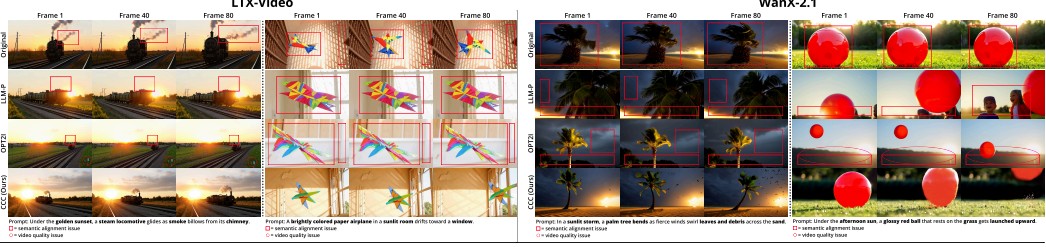

Figure 9: **Human preference scores (CCC vs. Origin) on Qwen-2.5-7b.** CCC consistently improves video generation quality across different video generation models.

### 4.5 RESULTS ON OTHER MODELS

To verify that CCC generalizes beyond o1 backbones, we applied the exact same loop (same $M = 3$ sampling budget, $F = 8$ frames). As Figure 9 shows, CCC delivers consistent gains over raw generation ("Origin"). This confirms that our multi-round critique-and-rewrite loop is broadly effective, regardless of the underlying text-to-video backbone.

## 5 CONCLUSION

We propose Critique Coach Calibration (CCC), a lightweight loop that lets a frozen text-to-video model learn from an off-the-shelf multimodal LLM: the LLM first critiques a generated clip, then rewrites the prompt, and the cycle repeats. Across two backbones, CCC lifts human-rated semantic alignment and visual quality significantly, without retraining or heavy post-editing. The method is transparent, cheap to deploy, and ablation studies show each design choice matters. We hope CCC encourages broader closed-loop generation systems that let evaluators directly coach generators.

**Ethics statement** The authors confirm that we have carefully reviewed and adhered to the ICLR Code of Ethics in this work.

**Reproducibility statement** We will open-source our codebase to ensure reproducibility. All the code, prompts, evaluation process, and generated videos during the experiment process will be released.

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

## A    LLM USAGE STATEMENT

The large language model is used during the manual script proofreading to detect grammatical errors and rephrase some overly long sentences. As mentioned in the paper, MLLM is also used to provide feedback for generated video.

## B    MLLM PROMPTS

We designed three closely related prompt templates to elicit both an error diagnosis and a corrective rewrite from the critic LLM. In each case, the prompt begins by identifying the input as an AI-generated video and including the original generation prompt (current_prompt). The first template reads:

```
This is a video generated by a video generation model.  The video is
generated by prompt: '<current_prompt>'. Please give a simple description
of the issue in this video, and also give a new prompt to generate a new
video to avoid the issue. The new prompt should not miss any details in
the original prompt. Please return the description in JSON format.  The
JSON format should be like this: {'issue': 'description', 'new_prompt':
'new_prompt'}.
```

We then extended this to explicitly acknowledge potential generation artifacts by adding the clause "Since it is a generated video, so there could be some issues," yielding the second template:

```
This is a video generated by a video generation model.  The video is
generated by prompt: '<current_prompt>'. Since it is a generated video,
so there could be some issues. Please give a simple description of the
issue in this video, and also give a new prompt to generate a new video to
avoid the issue. Do not miss any details in the original prompt. Please
return the description in JSON format.  The JSON format should be like
this: {'issue': 'description', 'new_prompt': 'new_prompt'}.
```

Finally, we introduced a quality rating field to capture coarse perceptual quality alongside the issue and rewrite, resulting in the third template:

```
This is a video generated by a video generation model.  The video is
generated by prompt: '<current_prompt>'. Since it is a generated video,
so there could be some issues. Please give a simple description of the
issue in this video, and also give a new prompt to generate a new video to
avoid the issue. Do not miss any details in the original prompt. Please
return the description in JSON format.  The JSON format should be like
this: {'quality': '1. Excellent, 2. Good, 3. Fair, 4. Poor, 5. Bad',
'issue': 'description', 'new_prompt': 'new_prompt'}.
```

For consistency with human evaluation scales, we interpret the quality field according to the ordered ratings "1. Excellent, 2. Good, 3. Fair, 4. Poor, 5. Bad." All three templates were applied to every video in our corpus to compare how explicit mention of potential issues and inclusion of a quality rating affect the critic's diagnostic output and the subsequent prompt refinement.

## C  QUALITATIVE EXAMPLES

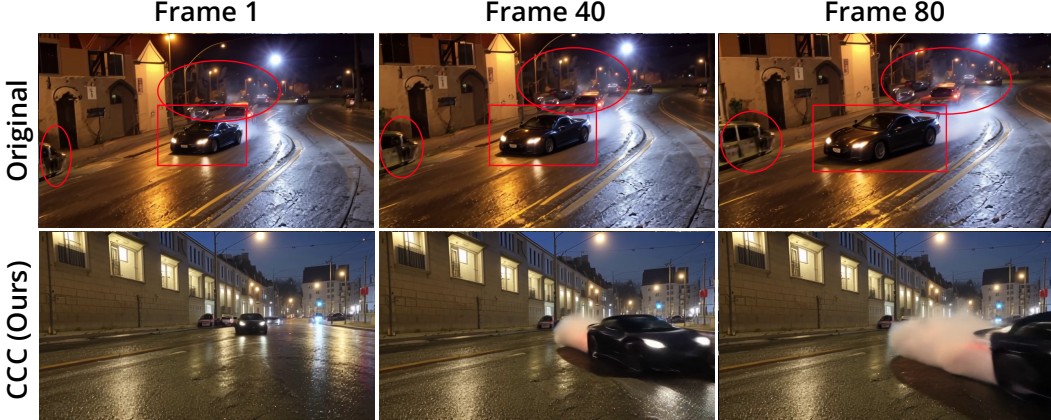

Prompt: Under **dim streetlights**, a **black sports car** expertly **drifts around a corner** with **trailing smoke**.
☐ = semantic alignment issue
◯ = video quality issue

Figure 10: **Comparison 1:** original vs. CCC-refined video. Each row shows frames 1/40/80 from the original (top) and our method (bottom), with semantic-alignment issues marked in red squares and video-quality issues in red circles.

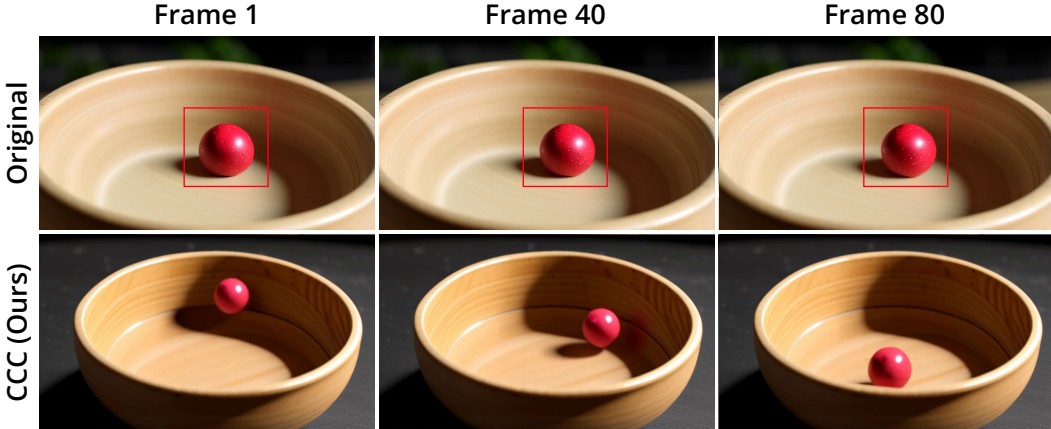

Prompt: In the **evening light**, a **small red ball** softly **rolls** around a **wooden bowl.**
☐ = semantic alignment issue
◯ = video quality issue

Figure 11: **Comparison 2:** original vs. CCC-refined video. Each row shows frames 1/40/80 from the original (top) and our method (bottom), with semantic-alignment issues marked in red squares and video-quality issues in red circles.

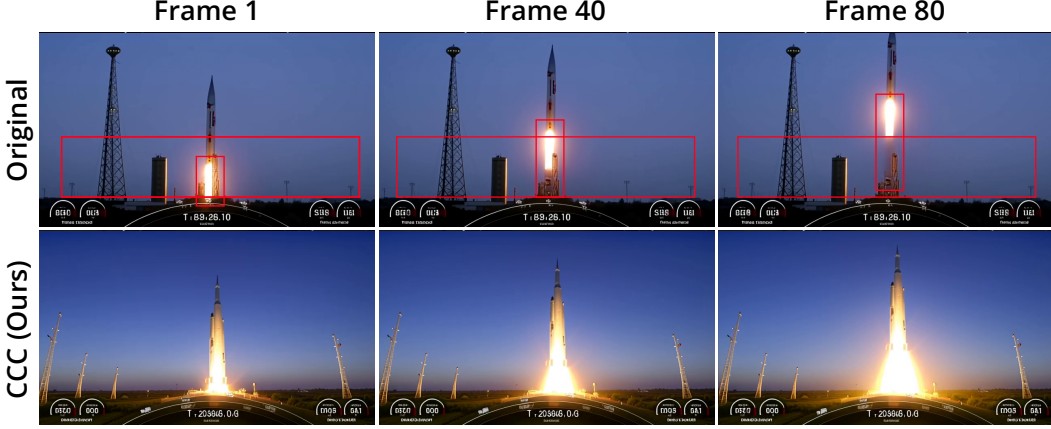

Prompt: At **sunset**, a **neon-green cyclist** smoothly **rounds the corner** as **spectators** cheer from behind.
□ = semantic alignment issue
○ = video quality issue

Figure 12: **Comparison 3:** original vs. CCC-refined video. Each row shows frames 1/40/80 from the original (top) and our method (bottom), with semantic-alignment issues marked in red squares and video-quality issues in red circles.

Prompt: Before **dawn**, a **rocket ignites**, **lighting the sky** with a fiery glow as it **ascends into space**.
□ = semantic alignment issue
○ = video quality issue

Figure 13: **Comparison 4:** original vs. CCC-refined video. Each row shows frames 1/40/80 from the original (top) and our method (bottom), with semantic-alignment issues marked in red squares and video-quality issues in red circles.

## D  IMPLEMENTATION DETAILS

Our entire pipeline is organized into three conceptual stages:

**Frame Sampling and Encoding.**  Each input video is first sampled to extract a fixed number of evenly spaced key frames. These frames are prepared for the critic LLM by converting them into a format suitable for multimodal input.

**LLM Critique Interface.**  We wrap all interactions with the frozen multimodal LLM (GPT-4o) in a lightweight interface that: (1) fills one of the predefined prompt templates with the current generation prompt and sampled frames; (2) issues multiple independent calls under identical inputs to obtain a set of critiques; and (3) parses the returned JSON or structured text into categorical labels and issue lists for downstream processing.

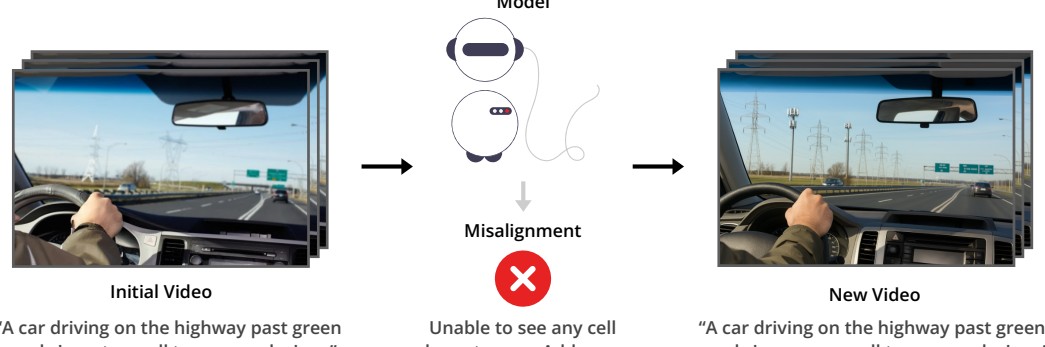

**Initial Video**

"A car driving on the highway past green road signs, two cell towers, and wires."

**Model**

**Misalignment**

Unable to see any cell phone towers. Add more.

**New Video**

"A car driving on the highway past green road signs, seven cell towers, and wires."

Figure 14: **Failure Case 1: Undetected objects trigger escalating hallucination.** When the critic fails to see the small towers in the initial clip, it requests to "add more," causing the refined prompt to overshoot from two to seven towers in the next iteration.

**Critique Coach Calibration Loop.** The core loop (Alg. 1) alternates between *critique* and *rewrite* phases. In the critique phase, we aggregate repeated LLM outputs to compute both a confidence score (self-consistency) and a content agreement score (variance analysis). In the rewrite phase, a second prompt template fuses the canonical critique back with the original prompt to produce a refined description. This two-stage cycle is repeated for a fixed number of iterations, yielding progressively improved prompts.

All configuration—number of sampled frames, self-consistency repetitions, iteration count, and choice of prompt templates—is exposed as a simple set of parameters. This modular design allows straightforward swapping of alternative prompt styles or LLM backends without touching the core orchestration logic.

## E    BASELINE IMPLEMENTATION DETAILS

We benchmark CCC against three *training-free* prompt-refinement baselines that keep **every** non-language-model hyper-parameter fixed—namely the frozen generator $G_\theta$, the clip budget $(M)$, the key-frame count $(F)$, and the GPT judge configuration described in Section 4.1. *Origin* forwards the raw prompt $P^{(0)}$ to $G_\theta$ and selects the best-rated clip, providing a no-feedback reference. *LLM-P* performs a single text-only paraphrase: GPT-4o rewrites $P^{(0)}$ *without* seeing any video, after which $M$ new clips are generated and scored. *OPT2I* executes one round of visual feedback: GPT-4o first paraphrases $P^{(0)}$, inspects a draft clip, issues a second rewrite $\hat{P}$, and we regenerate $M$ clips from $G_\theta(\hat{P})$. Because these variants differ only in the amount of critique and visual context (none, text-only, or one visual round), the gaps to CCC reported in Figure 4 isolate the value of our multi-round critique-and-rewrite loop.

## F    LIMITATION

Although Critique Coach Calibration can substantially boost model performance, it has limitations. Its agentic process involves repeated video model inference, leading to higher computational cost. In future work, we aim to reduce the required iterations.

## G    FAILURE CASES

Even though our evaluation-refinement loop is very successful, there are a few problems that deserve a deeper study.

**Low-quality input can nudge the loop off track (Fig. 14).** Compression artifacts, motion blur or noise may obscure small details (e.g. distant towers or wires) so the critic misreports missing

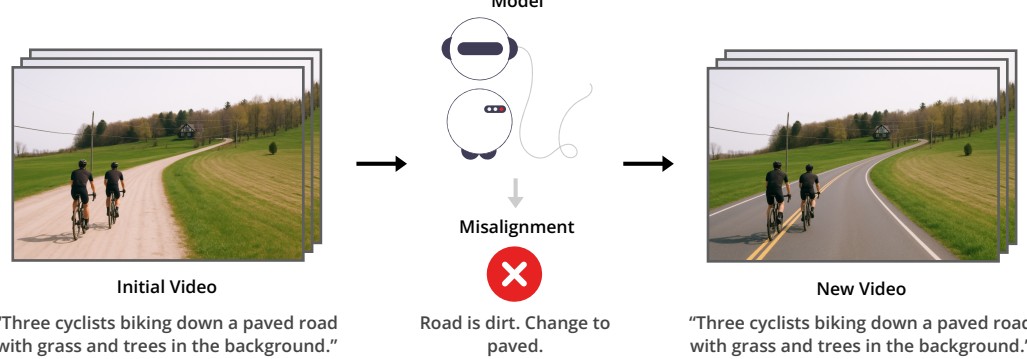

Figure 15: **Failure Case 2: A mislabeled surface type drives cascading feedback errors.** The ground-truth scene contains a dirt path, yet the prompt inaccurately calls it paved. Trusting this label, the critic instructs the generator to add pavement, so the refinement step paves the scene and the video drifts further from reality.

elements and the prompt rewriter "chases" phantom objects in subsequent iterations. A lightweight prefilter (e.g. denoising or super-resolution) or a confidence flag for low-quality frames would help prevent this mild but repetitive drift.

**Prompt–video mismatches amplify over rounds (Fig. 15).** If the initial prompt mislabels a scene element (e.g. calling dirt "paved") the critic dutifully enforces that error, and the loop compounds the mistake in later clips. While this usually causes only gradual content drift, integrating a simple verifier to check object counts or action labels before rewriting could stop propagation of obvious prompt errors.

IMPACT STATEMENT

This paper introduces `Critique Coach Calibration`, an automated, model-agnostic loop that iteratively critiques and rewrites prompts for text-to-video models, improving general issues found in the videos. The approach can benefit educational media, bias-controlled synthetic datasets for vision tasks.

