# OpenReview forum: "CCC: Prompt Evolution for Video Generation via Structured MLLM Feedback"
_ICLR.cc/2026/Conference — ICLR 2026 Conference Withdrawn Submission_

### Official Review · Reviewer_RKtz · 2025-10-30

**Soundness:** 2
**Presentation:** 2
**Contribution:** 1
**Rating:** 2
**Confidence:** 4

**Summary:**

This paper proposes CCC, a training-free, model-agnostic framework that leverages multimodal large language models (MLLMs) to iteratively critique and rewrite input text prompts for video generation. CCC aims to improve semantic alignment and perceptual quality by alternately diagnosing and updating prompts with the help of MLLM. Experiments are conducted on LTX-Video and WanX-2.1 backbones, showing gains in human visual preference and automatic assessment.

**Strengths:**

1. The paper aims at addressing a challenging issue in text-to-video generation -- poor semantic alignment between user prompt and generated video content -- which remains unsolved despite recent advancement in the video generation field. The idea of leveraging MLLM feedback to improve generation quality is technically sound.
2. The CCC framework (critic–coach loop) is intuitive and explained clearly.
3. The framework does not require fine-tuning of the MLLM and underlying video generator, which makes it lightweight and applicable to a range of existing models.

**Weaknesses:**

1. **Lack of Technical Novelty or Depth**. The proposed framework is largely an engineering workflow rather than a technically innovative algorithm. The core operation is simply prompting GPT-4o to critique and rewrite/refine text prompts. There is no new module, optimization, or theoretical insight. The contributions are incremental relative to prior “prompt-optimization via (M)LLM feedback” works (e.g., Idea2Img, OPT2I).
2. **MIssing Comparisons**. The is no comparison against more robust prompt-optimization methods or modern reinforcement feedback approaches (e.g., DPO-Video). and repairing methods (e.g., VideoRepair). Being training-free is a merit, but I am still curious about what the boundary of this line of work is and how they compare with RL-based methods.
3. **Insufficient Qualitative Evidence and Missing Videos**. The paper provides very few qualitative examples and no accompanying video results. Without visual evidence, it is impossible to verify the claimed improvements (aligment and perceptual quality).
4. **Marginal Improvement**. The proposed method cannot fundamentally improve the model’s capability -- for example, if the base model is unable to generate certain content, this approach cannot help. In contrast, a more effective prompt-extension method might achieve better alignment between user intent and the generated results.

**Questions:**

**Unclear Definition of Structured Feedback**. Although the paper claims “structured MLLM feedback,” the actual structure appears to be simple JSON outputs. Is there any verification that this structure contributes to stability or interpretability beyond ad-hoc formatting?

---

### Official Review · Reviewer_cH1n · 2025-10-30

**Soundness:** 2
**Presentation:** 3
**Contribution:** 2
**Rating:** 4
**Confidence:** 4

**Summary:**

This paper proposes Critique Coach Calibration (CCC), a training-free, model-agnostic framework for improving text-to-video generation.
CCC wraps a frozen generator in a self-diagnosing, self-correcting loop, where a multimodal LLM critiques generated videos and rewrites prompts accordingly.
Iterative refinement gradually enhances text–video alignment and perceptual quality.
Experiments on LTX-Video and WanX-2.1 demonstrate modest improvements in semantic fidelity and human preference, supported by ablation studies.

**Strengths:**

- Creative feedback formulation: CCC presents a closed feedback loop that connects generation and evaluation using an LLM as both critic and coach.

- Training-free and model-agnostic: The method requires no finetuning, weight updates, or extra models, making it simple to integrate into existing text-to-video pipelines.

**Weaknesses:**

- Limited novelty and theoretical depth: The core idea—using generated outputs’ deficiencies as feedback to refine prompts—has already been explored in the text-to-image domain, such as VisualPrompter (Wu et al., 2025) and Idea2Img (Yang et al., 2024).
CCC mainly extends this paradigm to video generation, summarizing it as a unified CCC pipeline.
The paper would benefit from clarifying what is conceptually new beyond this domain transfer—e.g., whether there is any new theoretical insight, evaluation principle, or temporal-consistency mechanism that specifically addresses video-specific challenges.

  - Wu, S., Sun, M., Wang, W., Wang, Y., & Liu, J. (2025). VisualPrompter: Prompt Optimization with Visual Feedback for Text-to-Image Synthesis. arXiv preprint arXiv:2506.23138.

  - Yang, Z., Wang, J., Li, L., Lin, K., Lin, C. C., Liu, Z., & Wang, L. (2024, September). Idea2img: Iterative self-refinement with gpt-4v for automatic image design and generation. In European Conference on Computer Vision (pp. 167-184). Cham: Springer Nature Switzerland.

- High computational overhead: Each CCC iteration requires two full rounds of video inference (generation + re-generation).
This substantially increases latency and GPU cost, making the framework less practical for real-world deployment or interactive systems.
The paper should provide a detailed trade-off analysis between computational cost and the quality improvements observed.

**Questions:**

- Could the authors explicitly position CCC against VisualPrompter (Wu et al., 2025) and Idea2Img (Yang et al., 2024), clarifying how CCC differs beyond domain extension?

- How does CCC balance quality improvement versus increased inference time? Any quantitative cost–benefit analysis?

- Does iterative rewriting risk semantic drift, gradually deviating from the original story?

- Is CCC sensitive to the choice of LLM (e.g., GPT-4o vs Gemini 2.5 Pro)?

---

### Official Review · Reviewer_mrpL · 2025-10-31

**Soundness:** 2
**Presentation:** 2
**Contribution:** 2
**Rating:** 4
**Confidence:** 3

**Summary:**

Current text-to-video generative models frequently generate videos that don't match the prompts. Existing solutions are either costly or introduce new problems. This paper's CCC proposes a method that iteratively refines prompts instead of modifying the video generation model. A multimodal language model critiques the generated videos, and the process of rewriting prompts based on that feedback is repeated, improving video quality and prompt alignment of existing models without additional training.

**Strengths:**

- The direction of finding better text prompts for video generation is promising.
- The design of the critics in the proposed approach is reasonable.
- Extensive experimental results are provided and each component is well ablated.

**Weaknesses:**

- The video generated at each iteration only depends on the improved prompt and does not reference previously generated videos. There is concern that this may be inefficient, as videos generated from improved prompts are newly created rather than improvements of existing videos.
- Given that current video generation models often fail to sufficiently follow prompts even with ideal video prompts, refining prompts may not be a robust approach to consistently refining videos.
- It is difficult to find information about which evaluation dataset was used and how many videos were generated and evaluated.
- The absence of video results makes the qualitative results less convincing.

**Questions:**

- It would be good if qualitative results showing how videos improve at each iteration can be provided.
- What are the video results (of each iteration step) when the number of iterations is greater than 3, for example 10?

---

### Official Review · Reviewer_nKPC · 2025-11-01

**Soundness:** 2
**Presentation:** 3
**Contribution:** 2
**Rating:** 4
**Confidence:** 4

**Summary:**

The paper proposes CCC, a training-free framework that iteratively improves text-to-video generation by using a MLLM to critique generated videos and refine prompts. This critique–coach loop iteratively uses a MLLM to score generated videos, gradually selecting optimized prompts, resampling videos, and choosing higher-quality video outputs. This paper conducts both qualitative and quantitative experiments to demonstrate that this method can improve the quality of generated videos to a certain extent.

**Strengths:**

1. The proposed method can be directly applied without requiring additional training data or fine-tuning.
2. The structure of this paper is clear, and the arrangement of the sections is reasonable.

**Weaknesses:**

1. This method requires multiple iterative samplings of several videos, which results in relatively low practical runtime efficiency.
2. Simply correcting the video by modifying prompts can only address misalignment between the generated video and the text. It is difficult to improve video fidelity or other quality aspects, which limits the further applicability of this method.
3. As a video generation work, more video results should be provided instead of presenting results as images in the paper. Using only images makes it hard to assess video fidelity, temporal consistency, and other quality metrics.

**Questions:**

Although MLLMs have developed rapidly, perceiving motion in videos remains challenging. Can MLLMs themselves accurately and clearly perceive dynamic content in videos, such as temporal consistency, video fidelity, and adherence to physical laws?

---

### Note · Authors · 2026-01-03

I have read and agree with the venue's withdrawal policy on behalf of myself and my co-authors.